# Transferless Inverted Graphene/Silicon Heterostructures Prepared by Plasma-Enhanced Chemical Vapor Deposition of Amorphous Silicon on CVD Graphene

**DOI:** 10.3390/nano10030589

**Published:** 2020-03-24

**Authors:** Martin Müller, Milan Bouša, Zdeňka Hájková, Martin Ledinský, Antonín Fejfar, Karolina Drogowska-Horná, Martin Kalbáč, Otakar Frank

**Affiliations:** 1Institute of Physics, Czech Academy of Sciences, Cukrovarnická 10, 162 00 Prague, Czech Republic; mullerm@fzu.cz (M.M.); hajkovaz@fzu.cz (Z.H.); ledinsky@fzu.cz (M.L.); fejfar@fzu.cz (A.F.); 2J. Heyrovský Institute of Physical Chemistry, Czech Academy of Sciences, Dolejškova 2155/3, 182 23 Prague, Czech Republic; milan.bousa@jh-inst.cas.cz (M.B.); karolina.drogowska@jh-inst.cas.cz (K.D.-H.); martin.kalbac@jh-inst.cas.cz (M.K.)

**Keywords:** silicon, graphene, heterostructure, CVD

## Abstract

The heterostructures of two-dimensional (2D) and three-dimensional (3D) materials represent one of the focal points of current nanotechnology research and development. From an application perspective, the possibility of a direct integration of active 2D layers with exceptional optoelectronic and mechanical properties into the existing semiconductor manufacturing processes is extremely appealing. However, for this purpose, 2D materials should ideally be grown directly on 3D substrates to avoid the transferring step, which induces damage and contamination of the 2D layer. Alternatively, when such an approach is difficult—as is the case of graphene on noncatalytic substrates such as Si—inverted structures can be created, where the 3D material is deposited onto the 2D substrate. In the present work, we investigated the possibility of using plasma-enhanced chemical vapor deposition (PECVD) to deposit amorphous hydrogenated Si (a-Si:H) onto graphene resting on a catalytic copper foil. The resulting stacks created at different Si deposition temperatures were investigated by the combination of Raman spectroscopy (to quantify the damage and to estimate the change in resistivity of graphene), temperature-dependent dark conductivity, and constant photocurrent measurements (to monitor the changes in the electronic properties of a-Si:H). The results indicate that the optimum is 100 °C deposition temperature, where the graphene still retains most of its properties and the a-Si:H layer presents high-quality, device-ready characteristics.

## 1. Introduction

The research on 2D materials, first of all on graphene (Gr), belongs to the most exciting areas in condensed matter physics. The possibility of realizing various heterostructures based on the combination of atomically thin layers with 3D counterparts offers a particularly encouraging playground to investigate and modulate electronic or optical properties. The combination of graphene (2D) with silicon (3D) has been intensively studied recently [1,2,3,4,5,6,7,8,9,10,11,12,13,14,15,16], as the formed graphene/silicon Schottky heterojunctions are believed to provide low-cost, ultrathin, and efficient electronic devices—for example, photodetectors or solar cells.

To realize a graphene/silicon heterostructure, chemical vapor-deposited (CVD) graphene is usually grown on a metal catalyst (for example, Cu, Ni, or Pt foil) and then transferred to a target silicon substrate by a sacrificial polymer-assisted method. Unfortunately, corrugations and cracks are formed, and the graphene layer can also be contaminated (by etchant and polymer residues) during the transfer process [17]. Therefore, commonly used transfer techniques (both dry and wet) using graphene-support polymers are not befitting for the assembly of device-quality silicon/graphene heterostructures, in particular not for industrial applications.

In this study, we propose an inverted course of action to produce graphene/silicon heterostructures, where a 2D material serves as a substrate for a-Si:H deposition performed by a well-known plasma-enhanced chemical vapor deposition (PECVD) process. The idea of silicon film deposition on CVD graphene has already been verified by Arezki et al. [18] and Lupina et al. [19]. However, in these cases, the graphene layer was transferred by polymer-assisted technique to SiO${}_{2}$/Si substrates before the PECVD process started. In our work, to evade a dubious interface and transfer-induced impurities, the a-Si:H films were grown straight on the CVD graphene-coated Cu foils. We interrogate a broad temperature series of PECVD deposition process (25–260 °C) to identify an optimum, where the defect-creation in graphene is minimized and, at the same time, the electrical conductivity of a-Si:H is maintained. We demonstrate PECVD of amorphous silicon as a feasible pathway for the production of superior graphene/silicon heterostructures that are not affected by the graphene transfer procedure.

## 2. Materials and Methods

The graphene monolayer was synthesized on copper catalyst substrate using a low-pressure (47 Pa) CVD setup [11]. The Cu foil (7 × 2 cm${}^{2}$; Alfa Aesar) was first heated to 1273 K and annealed with the flow of H${}_{2}$/Ar mixture [50 standard cubic centimeter per min (sccm)] for 20 min. Afterwards, 30 sccm of methane, as a carbon precursor, was introduced in the chamber for the same time of 20 min. Finally, the sample was cooled down to the room temperature. The quality of graphene was checked in each experiment by Raman spectroscopy (e.g., Appendix A). The growth conditions specified above led to a continuous coverage of predominantly monolayer graphene with thicker adlayers of lateral dimensions usually not exceeding 2–3 μm [20]. Minor heterogeneities in the Raman peak positions (Appendix A) correspond to the roughness and polycrystalline nature of the Cu foil [21]. The Cu foil with graphene was then quickly transferred in air to the PECVD chamber for a-Si:H deposition.

The intrinsic a-Si:H layers (approximately 20-nm-thick) were directly deposited on the graphene-Cu substrates (1 × 1 cm${}^{2}$) by a conventional capacitively coupled PECVD operating at a plasma frequency of 40 MHz. The substrates were heated up to various temperatures ranging between 25 °C and 260 °C and exposed to a glow discharge plasma of high-purity silane (99.999%) and hydrogen (99.99999%) gas mixture with an RF power density of 0.05 Wcm${}^{-2}$ under a pressure of 70 Pa. The total gas flow was set to 48 sccm. The a-Si:H film thickness was established ex situ with a Tencor Alpha–step 100 profilometer on a sample prepared during the same deposition run, but on a glass substrate.

Raman spectroscopy measurements were conducted on a LabRAM HR spectrometer (Horiba Jobin-Yvon) equipped with an Olympus BX-41 microscope (100× objective, N.A. = 0.9) and with the laser spot not exceeding 1 μm in diameter. For the excitation, a 633-nm (1.96-eV) laser with its power kept below 1 mW was used. To gather statistically relevant information, Raman mapping was conducted on the area comprising 30–40 μm${}^{2}$ with 2-μm mapping step. The Raman spectrometer was calibrated using an external Si reference, namely, the F${}_{1g}$ line at 520.5 cm${}^{-1}$. All evaluated Raman bands (D, G, D’, and 2D) were fitted by Lorentzian lineshapes.

Constant photocurrent measurement (CPM) and the temperature dependence of the dark conductivity were investigated on Corning glass (C7059) substrates with coplanar electrodes in a homemade setup equipped with a Keithley 237 source-measure unit.

## 3. Results and Discussion

The synthesis of graphene–silicon heterostructures requires the deposition of a device-quality silicon film on a graphene layer, which can be done by PECVD. However, the PECVD deposition of the silicon thin film leads the graphene to be exposed to quite violent conditions, with elevated temperature and various plasma species.

The effects of the a-Si:H PECVD process on the graphene layer were examined by Raman spectroscopy (Figure 1). The Raman spectra of the graphene were acquired through a ∼20-nm-thick a-Si:H film, which is sufficiently thin to get a reliable signal from the graphene beneath it. All the spectra exhibited a broad Raman band around 2200 cm${}^{-1}$, which was attributed to the Si-H bond. Additionally, all the spectra showed the G (“graphitic”, at the frequency of ∼1580 cm${}^{-1}$ for suspended graphene, assigned to the phonon with E${}_{2g}$ symmetry at the Γ point) and 2D (second-order resonant process; dispersive, at ∼2660 cm${}^{-1}$ for 633-nm excitation) peaks, which are characteristic for a graphene monolayer [22,23]. Additionally, the spectra with the grown silicon layer exhibited the D (intravalley mode, dispersive, at ≈1330 cm${}^{-1}$ for 633-nm excitation) and D’ (intervalley, ≈1615 cm${}^{-1}$) peaks, which are connected with the breaking of the inner symmetry (defects) of the graphene lattice [22,23].

The level of structural disorder in graphene—expressed as the distance between the defects ($L_{D}$) or, inversely, the defect density ($n_{D}$)—can be quantified through the intensity ratio between the D and G bands ($I_{D}/I_{G}$) [24,25]:(1)$$L_{D}^{2}\;(\mathrm{nm}{}^{2})\;=\left(1.8\pm 0.5\right)\times 10^{-9}\lambda_{L}^{4}{\left(\frac{I_{D}}{I_{G}}\right)}^{-1},$$
where λ is the excitation wavelength. The relation between $L_{D}$ and $n_{D}$ can be approximated as $L_{D}\approx n_{D}^{-0.5}$. Note the difference between $L_{D}$ and the lateral domain size ($L_{a}$), which is used to quantify disorder in 3D materials such as graphite [26,27]. However, the proposed protocol breaks down when $L_{D}$ or $L_{A}$ drop under a certain value (≈2–3 nm) [25,26,27]. For graphite, a three-stage model (a so-called amorphization trajectory) of the transition from the sp${}^{2}$ (graphitic) to the sp${}^{3}$ (tetrahedral) hybridized carbon atoms was introduced and tested [27]. In stage 1, the number of carbon vacancies increases, which is accompanied by the appearance and intensity increase of the D and D’ bands, and a G band upshift. The defect density is directly proportional to $I_{D}/I_{G}$. In stage 2, the defects start to coalesce and a greater amount of sp${}^{3}$ defects is observed as well (up to 20% by the end of stage 2 in graphite [27]). As the number of ordered aromatic rings decreases, the D band intensity is lowered too. Therefore, the defect density is inversely proportional to $I_{D}/I_{G}$ in stage 2. While the relation $1/n_{D}\propto I_{D}/I_{G}$ was empirically established for graphite, there is no precise enumeration for graphene due to the complex effects of different kinds of defects (vacancy, edgelike, sp${}^{3}$) on the defect bands [25,28]. Besides the $I_{D}/I_{G}$ drop, the G band frequency ($\omega_{G}$) in graphene decreases in stage 2; however, this is only when the amount of sp${}^{3}$ defects increases, not when the coalescence of vacancies takes place. Therefore, it is safer to utilize the width of the G band ($\Gamma_{G}$, defined as full-width at half-maximum) along with $I_{D}/I_{G}$ to monitor the degree of disorder because the width will always increase regardless of the defect type [25].

Figure 1a shows the evolution of the Raman spectra of graphene on Cu as a function of temperature at which a-Si:H is deposited (the Raman spectra for the whole series are depicted in Appendix A). All the main peaks of graphene reflect the increasing level of disorder induced by the plasma deposition process: the D and D’ bands intensities increase, all the bands broaden, and the 2D band intensity decreases. The analysis of data obtained by Raman mapping shows the steady increase of $\Gamma_{G}$ and $\Gamma_{D}$, as is detailed in Figure 1b,c, respectively. The evolution of the $I_{D}/I_{G}$ parameter is plotted in Figure 1d, along with $n_{D}$ calculated according to Equation (Equation 1). As can be seen, the $I_{D}/I_{G}$ increases until the deposition temperature reaches 250–260 °C. According to Equation (Equation 1), at this temperature, $n_{D}$ amounts to ≈4.5±2.5 ×$10^{11}$ cm${}^{-2}$, corresponding to $L_{D}\approx 10.3\pm 1.6$ nm. While the $L_{D}$ value still points to stage 1 amorphization, all the fitted parameters abruptly increase, especially the widths (see Figure 1b,c). In this case, the broadening of the bands is caused in part by the increasing disorder and in part by the more-pronounced heterogeneity. The inset in Figure 1b shows that the G band significantly downshifts in many of the mapped points for the deposition temperature of 260 °C. Such behavior is indicative of a large number of sp${}^{3}$ defects [27]. A certain degree of heterogeneity of the plasma-induced effects on graphene can be expected given the known variations of the reactivity of graphene depending on the Cu face it is resting on [29].

A deeper insight into the nature of the defects can be gained by looking at the ratio between the Raman intensities of the D and D’ bands ($I_{D}/I_{D^{\prime }}$) that reflects the nature of the defects [28]. In all tested samples, where the D’ can be confidently fitted (i.e., starting at 125 °C), the median $I_{D}/I_{D^{\prime }}$ varies in the range of 2.0–3.8. Even though there is a steady increase of the values with temperature, the range is indicative of dominantly edgelike defects [28]. It can be surmised that the reactive species in the plasma first attack the lower energy sites at the existing edges and grain boundaries, which are known to be more reactive [30,31], thereby extending them. However, at 260 °C, the spread of $I_{D}/I_{D^{\prime }}$ values increases rapidly, reaching even up to 8–9 in some cases, pointing to the appearance of vacancies or sp${}^{3}$ defects [28]. This corresponds to the observation of the G band downshift at 260 °C (see above).

It is known that the increasing disorder has adverse effects on the resistivity (ρ) of graphene. Due to the nature of our experiment, it was not possible to measure the electrical properties of the graphene itself because the particular level and type of disorder is achieved only after the deposition of Si, and its presence would, in turn, influence the measurement. However, there are numerous reports on the relation between $I_{D}/I_{G}$ (or $L_{D}$, $n_{D}$) and ρ (or sheet resistance, $R_{S}$) [32,33]. In stage 1, ρ ranges from ≈600 Ω of pristine CVD graphene to ≈ 20 kΩ [33]. For the $L_{D}$ range in our experiment, ρ should be lower than 1 kΩ at a 100-°C deposition temperature (with $L_{D}=24.1\pm 3.1$ nm), and it should reach ≈ 3 kΩ at 250 °C (with $L_{D}=9.8\pm 1.5$ nm).

The state of graphene in terms of charge-transfer doping and strain can also be evaluated from the correlation of G and 2D frequencies. The method was introduced by Lee et al. [34] and has been utilized in various studies and settings ever since [35], including on graphene on Si/SiO${}_{2}$ with a: Si-H deposited on top [19]. In brief, all the Raman data points (for example, from a map) are plotted in the $\omega_{2D}$, $\omega_{G}$ phase space. Due to the different sensitivity of the G and 2D bands to strain and charge transfer, a secondary coordinate system is created with the origin estimated from suspended graphene and the axes generated from the benchmark experiments on graphene deformation and doping. The $\omega_{2D}$ versus $\omega_{G}$ plots for the whole temperature series in our study are shown in Appendix A. Up to ≈200 C, the data points are spread in a mostly linear fashion in the $\omega_{2D}$, $\omega_{G}$ plots; however, the slope of the line (fitted by least squares) gradually decreases from ≈2.2 for bare graphene and, with a: Si-H deposited at 100 °C, down to ≈1.3 at 215 °C. The largest slope corresponds to the distribution of the data points only due to varying strain; the local charge carrier concentration does not significantly fluctuate, as is common for graphene on Cu [21]. A decreasing slope is indicative of the increasing influence of charge-transfer doping, related to the defect formation. At the highest temperatures, the distribution of the data points forms larger spreads in all directions, reflecting the great heterogeneity and high disorder in the lattice. The change in the $\omega_{2D}$, $\omega_{G}$ distribution for the as-grown graphene and graphene with a-Si:H deposited at 100 °C is detailed in Figure 1e. The clouds of the data points shift only along the isodoping line; in other words, only the strain is changing. The difference between the median values of the two distributions corresponds to a biaxial compression of ≈0.07%. In contrast to previous results of Si deposited on a graphene transferred to a Si/SiO${}_{2}$ substrate [19], no change in doping was observed at this deposition temperature. We might ascribe the difference to the state of graphene before the Si deposition—without significant impurities in our case against the transferred graphene with possible remnants of the sacrificial polymer (in the case of Reference [19]).

The electronic quality of silicon films strongly depends on the fabrication conditions. To find the optimum temperature interval at which graphene and a-Si:H exhibit properties suitable for device implementation, a series of silicon films deposited at various temperatures was characterized by the temperature-dependent dark conductivity and the CPM. The silicon films are identical to those deposited on graphene; however, the deposition time had to be increased to obtain the roughly 500-nm-thick film needed for reliable electrical and optical measurements.

Figure 2a displays the Arrhenius plot of the temperature dependence of dark conductivity $\sigma_{d}$ of a-Si:H deposited at various temperatures on Corning glass substrates with two coplanar titanium electrodes separated by 1.6 mm. These data are fitted to a singly activated conductivity, $\sigma_{d}=\sigma_{0}\;exp\left(-E_{a}/kT\right)$, where *k* is the Boltzmann constant and $E_{a}$ is the activation energy for electrical conduction. With the increase in a-Si:H deposition temperature, room temperature $\sigma_{d}$ increases and $E_{a}$ decreases, as shown in Figure 2b. More precisely, three areas of dependence of $\sigma_{d}$ on the substrate temperature during the a-Si:H deposition can be observed. At first, the dark conductivity increases with the increase in deposition temperature from room temperature to 100 °C. Then, the conductivity levels off at a value of $\approx 10^{-9}$ Scm${}^{-1}$ for deposition temperatures in the range of 100–200 °C. Finally, the dark conductivity rises again with the increase in deposition temperature above 200 °C.

Figure 3 shows the CPM mid-gap absorption spectra and the absorption coefficient α for 1.2 eV [α(1.2 eV)] of a-Si:H films deposited at various temperatures. The value of α(1.2 eV) is directly proportional to the concentration of deep-defects in a-Si:H, which are identified as the unsaturated (dangling) Si bonds [36]. The spectra were calibrated by transmission and reflection measurements. The value of α(1.2 eV) was calculated from the linear fit of the absorption spectra in the range 1.1–1.3 eV. From the calibration experiments [37], we can assign the absorption coefficient α(1.2 eV) value of 1 cm${}^{-1}$ to the dangling-bond density in the range of 2.4–5.0 $\times \;10^{16}$ cm${}^{-3}$ [38]. These defects serve as recombination centers for charge carriers, therefore, their densities are critical for solar cell properties [39], namely, open-circuit voltage and fill factor [36]. The device-quality a-Si:H is commonly considered to have a dangling-bond density of the order of $10^{15}$–$10^{16}$ cm${}^{-3}$ [36]. Hence, ideally, α(1.2 eV) should be approaching 0.1 cm${}^{-1}$. The absorption edge sharpness is the second important factor from a photovoltaic point of view. It is usually described by the so-called Urbach energy [40]. This parameter directly determines minimal losses in the open-circuit voltage in the finalized solar cell [41].

As shown in Figure 3, the a-Si:H films deposited at temperatures below 100 °C show large deep-defect densities, as can be seen from the high mid-gap absorption. The silicon films deposited at temperatures at and above 100 °C exhibit similar Urbach energies ($E_{u}=50$ meV) and low deep-defect densities; however, the absorption coefficient α(1.2 eV) and thus the deep-defect density slowly increase again with the increase in the deposition temperature. As shown in Figure 3b, the optimum temperature of a-Si:H PECVD deposition can be found—from the viewpoint of dangling bond defect density—at 100 °C. At this temperature, the $\sigma_{d}$ and $E_{a}$ are $\approx 10^{-9}$
$\Omega^{-1}$cm${}^{-1}$ and 0.7 eV, respectively (Figure 2), evincing the high-quality a-Si:H too. With respect to the Raman spectroscopy investigation, which showed only a minor change in the structural and electronic properties between the as-grown graphene and graphene with a-Si:H deposited at 100 °C, it is obvious that this particular temperature represents the optimum under our experimental conditions.

## 4. Conclusions

The inverted heterostructure of silicon grown directly on graphene resting on the catalytic Cu foil holds great promise towards circumventing the disorder and impurities that are imposed on the graphene when using the common transfer procedures to place the graphene on top of the silicon. After a-Si:H deposition by the PECVD method, the thus induced changes in graphene were monitored by Raman spectroscopy, allowing a direct quantification of the defect density and relating the possible changes in resistivity. The electronic properties of a-Si:H were assessed by temperature-dependent dark conductivity and constant photocurrent measurements, from which the activation energy of electronic conduction and dangling-bond density can be derived, respectively. An optimum a-Si:H growth temperature of 100 °C permitted us to fabricate a device-quality inverted graphene/silicon stack with minor graphene disorder and good electronic properties of the a-Si:H film. To conclude, we validated the a-Si:H growth by PECVD as a suitable method for the production of inverted graphene/silicon heterostructures, which can be relevant not only for possible photovoltaic applications but also for the “silicon” industry in general.

## Figures and Tables

**Figure 1 nanomaterials-10-00589-f001:**
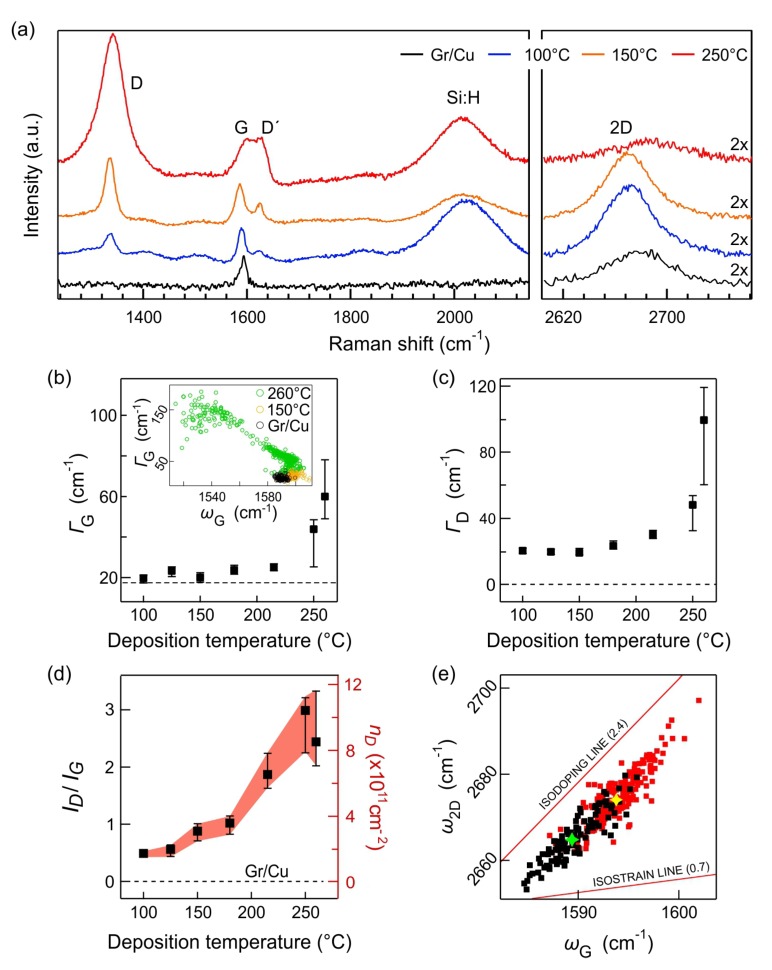
(**a**) Selected Raman spectra of as-grown graphene on the copper catalyst (black) and graphene with a-Si:H grown by plasma-enhanced chemical vapor deposition (PECVD) with deposition temperature in the range from 100 °C to 260 °C (colored). Raman 2D bands are multiplied 2× for clarity. Spectra are normalized on the Raman G band of graphene. (**b**,**c**) Evolution of $\Gamma_{G}$ and $\Gamma_{D}$, respectively, with increasing a-Si:H deposition temperatures. Inset in (b) shows correlation of $\Gamma_{G}$ with $\omega_{G}$ for selected deposition temperatures. (**d**) $I_{D}/I_{G}$ intensity ratios for increasing deposition temperature. Red-filled area indicates the estimated defect density according to Equation (Equation 1) (right axis). The spread of $n_{D}$ values is determined from the experimental error and the uncertainty from Equation (Equation 1). Horizontal dashed lines in (b–d) represent $\Gamma_{G}$, $\Gamma_{D}$, and $I_{D}/I_{G}$ median values for as-grown graphene; zero medians denote absence of the D band in the spectra. The data points in the main panels b–d represent medians of the fitted values from Raman maps comprising at least 225 points; the error bars are the first and third quartiles of the data distributions. (**e**) Correlation plot of the G and 2D frequencies for all points obtained during Raman mapping of the as-grown graphene (black squares) and graphene with a-Si:H deposited at 100 °C (red squares). Green and yellow asterisks mark medians of the datasets.

**Figure 2 nanomaterials-10-00589-f002:**
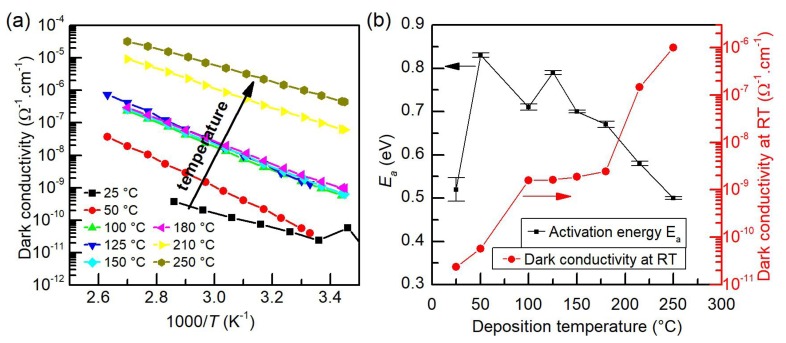
(**a**) Arrhenius plot of dark conductivity for a-Si:H films deposited at various temperatures in the range of 25 °C to 250 °C; and (**b**) the dependence of activation energy $E_{a}$ and room-temperature dark conductivity on the deposition temperature (the error bars represent the standard error of the linear regression of the data sets in (a)). The lines between the points are drawn to guide the eye.

**Figure 3 nanomaterials-10-00589-f003:**
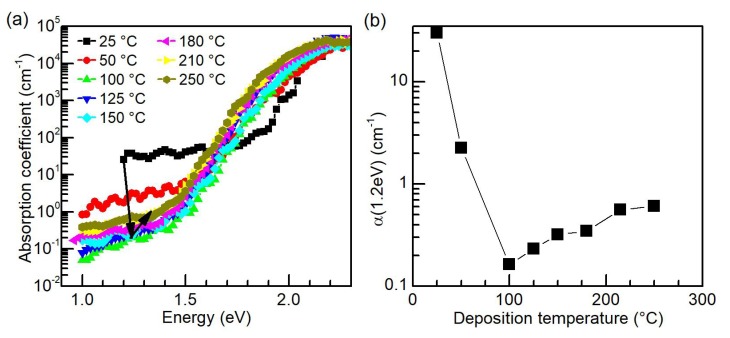
(**a**) Constant photocurrent measurement (CPM) measurement of mid-gap absorption of a-Si:H films deposited at various temperatures in the range of 25 °C to 250 °C; and (**b**) the dependence of the absorption coefficient α on the deposition temperature. The lines between the points are drawn to guide the eye.

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
