# Peer review of "Transferless Inverted Graphene/Silicon Heterostructures Prepared by Plasma-Enhanced Chemical Vapor Deposition of Amorphous Silicon on CVD Graphene"

_nanomaterials, 2020, doi:10.3390/nano10030589_

Round 1

Reviewer 1 Report

This manuscript reported the deposition of a-Si:H on a graphene-covered Cu foil using PECVD. It basically demonstrated that 3D Si can be grown on 2D graphene to form an inverted structure unlike the traditional approach. I thought it is an original work and provides a different thinking from most of the CVD research. This initial step may attract other scholars to pursue in this direction and finally get into some real application areas. Detailed Raman spectra, dark conductivity and constant photocurrent measurement results were presented and discussed. In my opinion, the results are positive with interesting inputs. In general, this paper is clear and well written. Also the conclusions are consistent with the experimental evidence. Hence, I recommend this manuscript to be published in Nanomaterials.

Author Response

We are thankful to the reviewer for the positive report.

Reviewer 2 Report

The manuscript reports on the fabrication of an inverted heterostructure of a-Si:H on graphene.

The manuscript is well written and experiments clearly described. There are only a few minor points that need to be addressed:

1) How the authors estimated the 20 nm thickness on graphene? If it is evaluated based on the deposition time, the authors should comment on how the thickness has been evaluated.

2) The authors report that the Cu foil was annealed with a flow of "H2[50 standard cubic cm (sccm)]/He". What is the meaning? Is the flow a mixture of H2 and He?

3) The  sccm should be correctly reported as standard cubic centimeter per min.

Overall, the manuscript is interesting and deserves publication after minor revision.

Author Response

1) How the authors estimated the 20 nm thickness on graphene? If it is evaluated based on the deposition time, the authors should comment on how the thickness has been evaluated.

> The a-Si:H film thickness was established ex-situ with a Tencor Alpha–step 100 profilometer on a sample prepared during the same deposition run, but on a glass substrate. We have clarified the detail in the Materials and Methods section.

2) The authors report that the Cu foil was annealed with a flow of "H2[50 standard cubic cm (sccm)]/He". What is the meaning? Is the flow a mixture of H2 and He?

> We are thankful for pointing out our mistake. The flow is a mixture of H2/Ar; we have corrected it.

3) The  sccm should be correctly reported as standard cubic centimeter per min.

> Corrected. Together with the correction from point 2 and a specification of the methane flow, the part of the sentence in Materials and Methods reads now: ‘…annealed with the flow of H2/Ar mixture [50 standard cubic centimeter per min (sccm)] for 20 minutes. Afterwards, 30 sccm of methane, as a carbon precursor,…’

Overall, the manuscript is interesting and deserves publication after minor revision.

> We are thankful to the reviewer for the positive report and the issues pointed out for clarification.

Reviewer 3 Report

In the manuscript entitled “Transfer-less inverted graphene/silicon heterostructures prepared by plasma-enhanced chemical vapor deposition of amorphous silicon on CVD graphene”, the authors proposed and investigated, principally by RAMAN spectroscopy, an alternative method to produce a graphene/silicon heterostructure by PECVD, attempting to reduce the defects on graphene typical of the conventional transfer method.

The procedure consists of two steps: 1) graphene was grown on a Cu substrate (which size?) by CVD; 2) silicon was grown on Gr/Cu by PECVD.

The quality of the Gr/Cu substrate has not been discussed accurately. How many layers of graphene were deposited? Was graphene homogeneous distributed on the Cu substrate?

Was Si deposited in situ or Gr/Cu was exposed to air before deposition?

Which kind of defects can be induced by the Si grown? Were just sp3 sites created? Or could be holes created?

Why the error bars shown in figure 1 are not centered on the points? And why their amplitude is different?

To perform the RAMAN spectroscopy investigations, the authors deposited just 20 nm of Si. But what does happen if a larger thickness is deposited?

Page 1 raw 19. “is considered one of the most promising…”, Gr can’t be considered promising , it’s rather consolidated topic. Please modify.

Page 5 raw 131. Change comma with point before “In brief…”

Author Response

> We are thankful to the reviewer for pointing to the non-clarified issues. Below is our point-by-point reply to the individual comments, incl. the main changes performed in the text.

The procedure consists of two steps: 1) graphene was grown on a Cu substrate (which size?) by CVD; 2) silicon was grown on Gr/Cu by PECVD.

> The initial size of the Cu foil for graphene growth was 7x2 cm2; which was then cut into pieces of 1x1 cm2 for the Si deposition. We have added this information to the Materials and Methods section.

b) The quality of the Gr/Cu substrate has not been discussed accurately. How many layers of graphene were deposited? Was graphene homogeneous distributed on the Cu substrate?

> We have added the following text and reference nr. 20 (Kalbac et al. Carbon 2012) to the first paragraph of Materials and Methods section to specify the quality of Gr on Cu substrate:

The quality of graphene was checked in each experiment by Raman spectroscopy [e.g., Figures S1 and S2, Supporting Information (SI)]. The growth conditions specified above lead to a continuous coverage of predominantly monolayer graphene with thicker adlayers of lateral dimensions usually not exceeding 2–3μm [20]. Minor heterogeneities in the Raman peak positions (Figure S2, SI) correspond to the roughness and polycrystalline nature of the Cu foil [21].

c) Was Si deposited in situ or Gr/Cu was exposed to air before deposition?

> The Gr/Cu was exposed to air during transfer between the growth chambers. The following sentence was added to the first paragraph of Materials and Methods section:

The Cu foil with graphene was then quickly transferred in air to the PECVD chamber for a-Si:H deposition.

d) Which kind of defects can be induced by the Si grown? Were just sp3 sites created? Or could be holes created?

> We can have only indirect insight into the nature of the defects, utilizing the Raman intensity D/D’ ratio as described by Eckmann et al. (2012). Upon analyzing the data, we have come to the conclusion that the majority of defects we observe can be edge-type, originated by extending the existing edges or domain boundaries, which are known to be more reactive. We have added the following paragraph and two references 30,31 (Girit et al. Science 2009, Bisset et al ACS Nano 2012) (page 5, lines 126-135):

A deeper insight into the nature of the defects can be gained by looking at the ratio between the Raman intensities of the D and D’ bands (ID/ID′) that reflects the nature of the defects [28].  In all tested samples, where the D’ can be confidently fitted (i.e. starting at 125◦C), the median ID/ID′ varies in the range of 2.0–3.8. Even though there is a steady increase of the values with temperature, the range is indicative of dominantly edge-like defects [28]. It can be surmised that the reactive species in the plasma attack first the lower energy sites at the existing edges and grain boundaries, which are known to be more reactive [30,31], and thereby extending them. However, at 260◦C, the spread of ID/ID′ values increases rapidly, reaching even up to 8–9 in some cases, pointing to the appearance of vacancies or sp3 defects [28]. This corresponds to the observation of the G band downshift at 260◦C (see above).

e) Why the error bars shown in figure 1 are not centered on the points? And why their amplitude is different?

> As specified in the Figure 1 caption, the data points in panels b-d represent median values obtained from Raman mapping with first and third quartiles as the error bars. The non-centered error bars and their different amplitudes reflect the deviations from normal distributions and the spread of the mapping data. To clarify the meaning of the data points in these panels, we have rephrased and unified the description in Figure 1 caption and added the following sentence:

The data points in the main panels b--d represent medians of the fitted values from Raman maps comprising at least 225 points; the error bars are the first and third quartiles of the data distributions.

f) To perform the RAMAN spectroscopy investigations, the authors deposited just 20 nm of Si. But what does happen if a larger thickness is deposited?

> We selected the mentioned thickness as a compromise between the continuous coverage of Si and the strength of the collected Raman signal given by the limited transparency of the deposited Si to the visible light. From a different experiment we know there is a larger heterogeneity of the Raman data if a thinner Si is employed, even though the same maximum D/G intensity ratio is achieved for a given temperature regardless the Si thickness. We thus do not expect the main features of the Raman spectra to change with the increasing Si thickness.

g) Page 1 raw 19. “is considered one of the most promising…”, Gr can’t be considered promising , it’s rather consolidated topic. Please modify.

> We have modified the sentence to read:

The research on 2D materials, first of all on graphene (Gr), belongs to the most exciting areas in condensed matter physics.

h) Page 5 raw 131. Change comma with point before “In brief…”

> Corrected.